# KATANIN-dependent mechanical properties of the stigmatic cell wall mediate the pollen tube path in Arabidopsis

Lucie Riglet[1†], Frédérique Rozier[1], Chie Kodera[1‡], Simone Bovio[1], Julien Sechet[2], Isabelle Fobis-Loisy[1], Thierry Gaude[1]*

[1]Laboratoire de Reproduction et Développement des Plantes, Université de Lyon, ENS de Lyon, UCBL, INRAE, Allée d'Italie, France; [2]Institut Jean-Pierre Bourgin, INRAE, AgroParisTech, Université Paris-Saclay, Versailles, France

**Abstract** Successful fertilization in angiosperms depends on the proper trajectory of pollen tubes through the pistil tissues to reach the ovules. Pollen tubes first grow within the cell wall of the papilla cells, applying pressure to the cell. Mechanical forces are known to play a major role in plant cell shape by controlling the orientation of cortical microtubules (CMTs), which in turn mediate deposition of cellulose microfibrils (CMFs). Here, by combining imaging, genetic and chemical approaches, we show that isotropic reorientation of CMTs and CMFs in aged Col-0 and *katanin1-5* (*ktn1-5*) papilla cells is accompanied by a tendency of pollen tubes to coil around the papillae. We show that this coiled phenotype is associated with specific mechanical properties of the cell walls that provide less resistance to pollen tube growth. Our results reveal an unexpected role for KTN1 in pollen tube guidance on the stigma by ensuring mechanical anisotropy of the papilla cell wall.

*For correspondence:
thierry.gaude@ens-lyon.fr

Present address: †Sainsbury Laboratory, University of Cambridge, Cambridge, United Kingdom; ‡Institut Jean-Pierre Bourgin, INRAE, AgroParisTech, Université Paris-Saclay, Versailles, France

Competing interests: The authors declare that no competing interests exist.

## Introduction

Following deposition of dehydrated pollen grains on the receptive surface of the female organ, the stigma, pollen rehydrates, germinates and produces a pollen tube that carries the male gametes toward the ovules where the double fertilization takes place. This long itinerary through the different tissues of the pistil is finely controlled, avoiding misrouting of the pollen tube and hence assuring proper delivery of the sperm cells to the female gametes. How the pollen germinates a tube and how pollen tube growth is regulated have been the object of many investigations (*Palanivelu and Tsukamoto, 2012*; *Mizuta and Higashiyama, 2018*). The use of in vitro pollen germination as well as semi-in vivo fertilization assays, together with the analysis of mutants defective in pollen or ovule functions, have rapidly expanded our knowledge of the mechanisms that sustain pollen tube growth, its guidance toward the ovule and the final delivery of male gametes within the embryo sac (*Dresselhaus and Franklin-Tong, 2013*; *Higashiyama and Yang, 2017*; *Cameron and Geitmann, 2018*). At the cellular level, cytoskeleton has been extensively studied during pollen tube elongation (*Fu, 2015*), highlighting the critical role played by the actin microfilaments in pollen-tube tip growth through delivery of materials for the biosynthesis of the plasma membrane and cell wall. By contrast, CMTs seem to have a lower importance in the pollen tube elongation process, drugs affecting CMT polymerisation having no significant effects on pollen-tube growth rate, yet altering the capacity of pollen tubes to change their growth direction (*Gossot and Geitmann, 2007*).

In *Arabidopsis thaliana*, pollen tubes grow within the cell wall of papillae of the stigmatic epidermis, and then through the transmitting tissue of the style and ovary (*Lennon and Lord, 2000*). The

**eLife digest** Flowering plants produce small particles known as pollen that – with the help of the wind, bees and other animals – carry male sex cells (sperm) to female sex cells (eggs) contained within flowers. When a grain of pollen lands on the female organ of a flower, called the pistil, it gives rise to a tube that grows through the pistil towards the egg cells at the base.

The surface of the pistil is covered in a layer of long cells named papillae. Like most plant cells, the papillae are surrounded by a rigid structure known as the cell wall, which is mainly composed of strands known as microfibrils. The pollen tube exerts pressure on a papilla to allow it to grow through the cell wall towards the base of the pistil. Previous studies have shown that the pistil produces signals that guide pollen tubes to the eggs. However, it remains unclear how pollen tubes orient themselves on the surface of papillae to grow in the right direction through the pistil.

Riglet et al. combined microscopy, genetic and chemical approaches to study how pollen tubes grow through the surface of the pistils of a small weed known as *Arabidopsis thaliana*. The experiments showed that an enzyme called KATANIN conferred mechanical properties to the cell walls of papillae that allowed pollen tubes to grow towards the egg cells, and also altered the orientation of the microfibrils in these cell walls. In *A. thaliana* plants that were genetically modified to lack KATANIN the pollen tubes coiled around the papillae and sometimes grew in the opposite direction to where the eggs were.

KATANIN is known to cut structural filaments inside the cells of plants, animals and most other living things. By revealing an additional role for KATANIN in regulating the mechanical properties of the papilla cell wall, these findings indicate this enzyme may also regulate the mechanical properties of cells involved in other biological processes.

transmitting tissue has an essential function in pollen tube guidance, providing chemical attractants and nutrients (*Crawford and Yanofsky, 2008*; *Higashiyama and Hamamura, 2008*). In contrast to these accumulating data showing the existence of factors mediating pollen tube growth in the pistil, whether guidance cues exist at the very early stage of pollen tube emergence and growth in the papilla cell wall remains largely unknown. The cell wall constitutes a stiff substrate and hence a mechanical barrier to pollen tube progression. There are numerous examples in animal cells demonstrating that mechanical properties of the cellular environment, and in particular rigidity, mediate cell signalling, proliferation, differentiation and migration (*Discher et al., 2005*; *Fu et al., 2010*; *Ermis et al., 2018*). In plant cells, cell wall rigidity depends mainly on its major component, cellulose, which is synthesized by plasma membrane-localized cellulose synthase complexes (CSCs) moving along cortical microtubule (CMT) tracks (*Paredez et al., 2006*). While penetrating the cell wall, the pollen tube exerts a pressure onto the stigmatic cell (*Sanati Nezhad and Geitmann, 2013*). Physical forces are known to reorganise the cortical microtubules (CMTs), which by directing CSCs to the plasma membrane, reinforce wall stiffness by novel cellulose microfibril (CMF) synthesis (*Paredez et al., 2006*; *Sampathkumar et al., 2014*). Hence, there is an intricate interconnection between CMT organisation, CMF deposition and cell wall rigidity (*Xiao and Anderson, 2016*). A major regulatory element of CMT dynamics is the KATANIN (KTN1) microtubule-severing enzyme, which allows CMT reorientation following mechanical stimulation (*Uyttewaal et al., 2012*; *Sampathkumar et al., 2014*; *Louveaux et al., 2016*). Here we investigated whether the CMT network of papilla cells might contribute to pollen tube growth and guidance in stigmatic cells by combining cell imaging techniques, genetic tools, chemical analysis of the cell wall and atomic force microscopy. We show that isotropic reorientation of CMTs occurs in aged Col-0 and *ktn1-5* papilla cells, which is accompanied by a change in the growth direction of pollen tubes that tend to make coils on papillae. We show that CMT reorganisation is associated with isotropic rearrangement of CMFs, high content of crystalline cellulose and particular cell wall mechanics of *ktn1-5* stigmas. Altogether, our results indicate that cytoskeleton dynamics and mechanical properties of the cell wall, which both depend on KTN1 activity, have a major role in guiding early pollen tube growth in stigma papillae.

## Results

### CMT dynamic pattern and pollen tube growth during stigma development

To assess the functional role of stigmatic CMTs in pollen-papilla cell interaction, we first analysed their organisation in papillae at stages 12 to 15 of stigma development as described (*Smyth et al., 1990*; *Figure 1A,B*). We generated a transgenic line expressing the CMT marker MAP65.1-citrine under the control of the stigma-specific promoter SLR1 (*Fobis-Loisy et al., 2007*). Before (stage 12) and at anthesis (stage 13), where flowers are fully fertile (*Kandasamy et al., 1994*), the CMTs were aligned perpendicularly to the longitudinal axis of papilla cells and were highly anisotropic (median value of 0.40) (*Figure 1C,D*). At stage 14, when anthers extend above the stigma, the CMT pattern became less organised, with a higher variability in anisotropy values. Finally, at stage 15 when the stigma extends above anthers, CMT anisotropy had a median value of 0.09 indicative of an isotropic orientation of CMTs (*Figure 1C,D*). These findings reveal that the papilla CMT cytoskeleton is dynamic during development, with a change of CMT array orientation from anisotropy to isotropy. We then wondered whether this change in CMT organisation could be correlated with pollen tube growth. To this end, we self-pollinated Col-0 papillae from stages 12 to 15 and examined pollen tube growth one hour after pollination by scanning electron microscopy (SEM) (*Figure 2A*). At stage 12 and 13, we found most (~60%) pollen tubes grew straight in the papillae, whereas about 30% and 10% of tubes made half-turn or one turn around stigmatic cells, respectively (*Figure 2B*). At later stages of development, the tendency to coil around the papillae increased, with 3.5-fold more pollen tubes at stage 14 and 5.5-fold more at stage 15 making one or more than one turn around papillae. These results suggest that CMT organisation in the papilla impacts the direction of growth of pollen tubes and that loss of CMT anisotropy is associated with coiled growth.

### Pollen tube penetration in the papilla cell wall does not alter CMT Organisation

We next asked whether the pressure exerted by the growing pollen tube in the cell wall may induce changes in the CMT organisation of the papilla cell. To this end, we set up a live imaging system where pollinated stigmas were

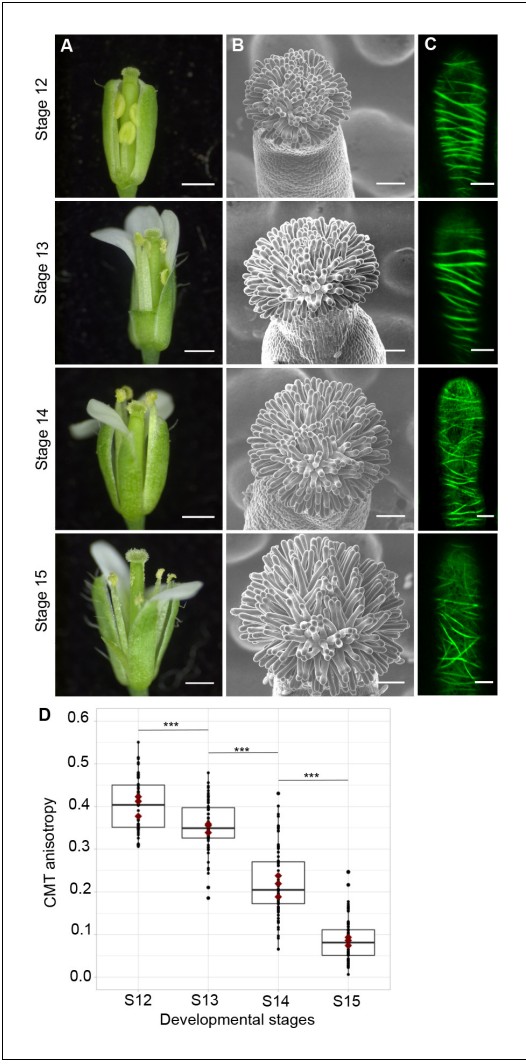

**Figure 1.** CMT organisation during papilla cell development. (**A**) Flower development of *A. thaliana* from developmental stages 12 to 15. Scale bar, 500 μm. (**B**) Upper view of the stigma during development by SEM. Scale bar, 50 μm. (**C**) Confocal images of papilla cells expressing MAP65-citrine at each stage of development. Scale bar, 5 μm. (**D**) Quantitative analysis of CMT array anisotropy of papilla cells from stages 12 to 15. The red dots correspond to the mean values of the three replicates. Statistical differences were calculated using a Shapiro-Wilk test to evaluate the normality and then a Wilcoxon test, ***p<0.01. N > 4 stigmas, n > 60 papilla cells for each stage. *Figure 1—source data 1* provides data for assessing CMT anisotropy shown in D.

The online version of this article includes the following source data for figure 1:

**Source data 1.** Source data for CMT organisation shown in *Figure 1D*.

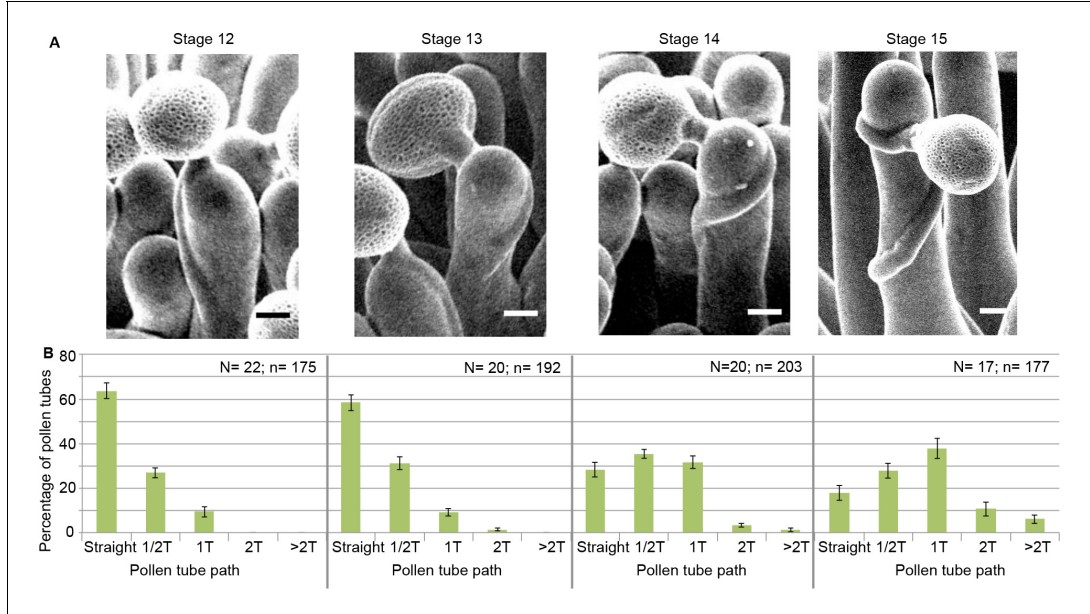

**Figure 2.** Pollen tube growth behaviour on papillae during development. (**A**) SEM images of Col-0 papillae pollinated with Col-0 pollen, one hour post pollination, from stages 12 to 15. Scale bar, 5 μm. (**B**) Quantification of the number of turns (**T**) made by the pollen tube on papillae from stages 12 to 15. Data are expressed as mean +/- s.e.m. A Chi-Square test for independence (at 6 degrees of freedom) was used to compare all stages and demonstrated that the number of turns was significantly different between stages (p<0.01). N corresponds to the number of stigmas analysed in this study and n, the number of papillae. *Figure 2—source data 1* provides data for the quantification of the number of turns made by Col-0 pollen tubes on papillae from stage 12 to 15 shown in **B**.

The online version of this article includes the following source data for figure 2:

**Source data 1.** Source data for the number of turns made by the pollen tube on papillae from stage 12 to 15 shown in *Figure 2B*.

maintained in air so as to prevent immediate hydration and burst of pollen grains. During the time course of the experiment, starting at the beginning of pollen tube emergence on the stigma papilla at stage 13 (t0) up to 13 min after germination, we found no major alteration of CMT arrays from early to late penetration stages (*Figure 3A*). This result was in contradiction with previous immunostaining analysis that showed fragmentation of CMTs following compatible pollination in *Brassica napus* (*Samuel et al., 2011*). To confirm our result and gain better insight into CMT organisation, we then observed pollinated stigmas at a single time point, that is 30 min after pollination, using a mounting medium to get higher resolution images. Again, we did not detect any reorganisation or degradation of CMTs (*Figure 3B*). However, we found the stability of CMTs to be very sensitive to strong pressure, as when the coverslip was applied not cautiously on the stigma, a break-down of CMT arrays occurred (*Figure 3—figure supplement 1*). These results indicate that the pressure exerted by the pollen tube while growing in the papilla cell wall does not alter the organisation of stigmatic CMTs.

## Impaired CMT dynamics of papillae affects pollen tube growth direction

To test the direct impact of stigma CMTs on pollen tube growth direction, we examined whether the destabilization of CMTs in Col-0 papillae could affect pollen tube growth. To this end, we treated stigmas by local application of the depolymerizing microtubule drug oryzalin in lanolin pasted around the style. After 4 hr of drug treatment, no more CMT labelling was detected in papillae, while CMTs were clearly visible in mock-treated (DMSO) stigmas (*Figure 4A*). Stigmas were then pollinated with Col-0 pollen and one hour later observed by SEM. Pollen tubes were found to make one turn more frequently on drug-treated than on control papillae, with more than a 3-fold increase of tubes making one turn (*Figure 4B,C*). To confirm the relation between stigma CMTs and pollen behaviour, we examined pollen tube growth on stigmas of the *katanin1-5* mutant, which is known to exhibit reduced CMT array anisotropy in root cells (*Bichet et al., 2001*; *Burk et al., 2001*; *Burk and*

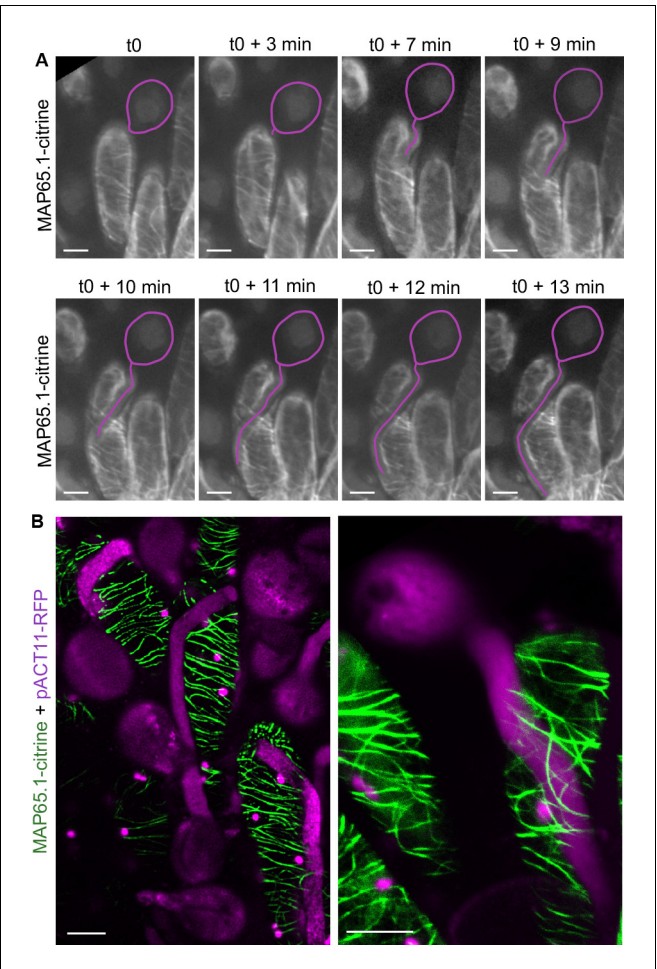

**Figure 3.** CMT organisation following pollen tube growth. (**A**) Time-lapse imaging of Col-0 pollen tube growth within a papilla cell expressing MAP65.1-citrine. Pollen outline and pollen tube path were highlighted in magenta for a better visualisation. N = 5 papillae. (**B**) Confocal images of papilla cells expressing MAP65.1-citrine (in green), 30 min after pollination with RFP labelled pollen grains (in magenta). The magenta dots are chloroplasts. Right image: close-up of a pollinated papilla, 3D projection. N = 13 papillae. Scale bars, 10 μm. *Figure 3—figure supplement 1* shows the destabilisation and fragmentation of CMTs upon mechanical stimulation.

The online version of this article includes the following figure supplement(s) for figure 3:

**Figure supplement 1.** Confocal image of the effect of mechanical stimulation on CMT organisation.

*Ye, 2002*). Because the CMT organisation in *ktn1-5* papillae is unknown, we crossed *ktn1-5* with the MAP65-1-citrine marker line and found that CMT arrays were more isotropic in *ktn1-5* papillae when compared with those of the WT (*Figure 5A,C*). We then analysed Col-0 pollen behaviour on *ktn1-5* stigmatic cells at stage 13 (*Figure 5B*). We observed that Col-0 pollen tubes acquired a strong tendency to coil around *ktn1-5* papillae, with more than 60% of tubes making one or more than one turn around papillae, sometimes making up to six turns, before reaching the base of the cell (*Figure 5D*; *Figure 5—figure supplement 1A–C*). In some rare cases, pollen tubes even grew upward in the *ktn1-5* mutant and appeared blocked at the tip of the papilla (*Figure 5—figure supplement 1B,D*). The number of coils was significantly higher than on oryzalin-treated papillae. Indeed, 25% of the pollen tubes made at least two coils in *ktn1-5* papillae whereas this percentage represented only 4% on the oryzalin treated stigmas (*Figure 4C*; *Figure 5D*). To check whether papilla receptivity to pollen might be impaired in *ktn1-5* stigmas, we examined hydration and tube growth of Col-0 pollen deposited on stage-13 Col-0 or *ktn1-5* stigmas. We found no significant differences in pollen hydration related to the stigma type (*Figure 5—figure supplement 2A*) and pollen tubes grew in the transmitting tract and reached the ovules in *ktn1-5* pistils like in Col-0

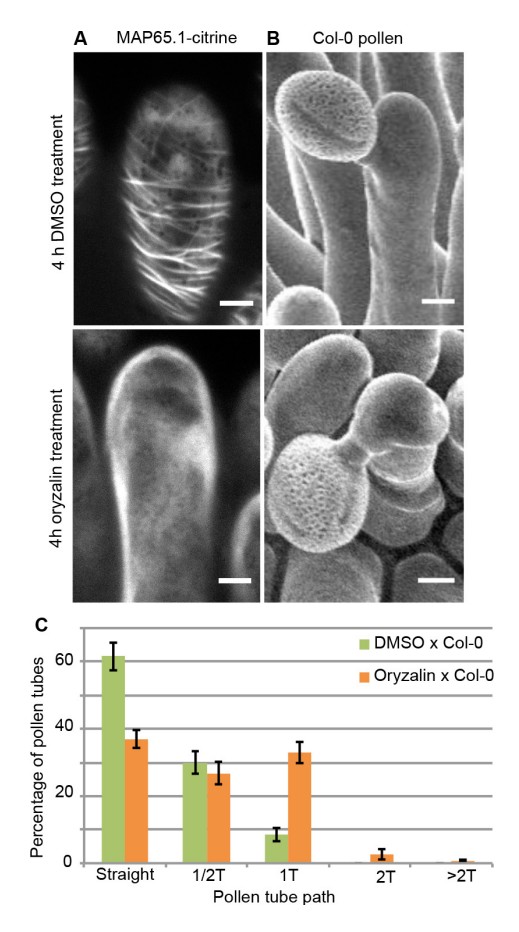

**Figure 4.** Local oryzalin application on Col-0 stigmas promotes MT destabilization and induces Col-0 pollen tube coils. (**A**) Col-0 papilla cells expressing MAP65.1-citrine after 4 hr of DMSO (top) or oryzalin (bottom) local treatment. (**B**) SEM images of DMSO- (top) or oryzalin-treated (bottom) Col-0 stigmas pollinated with Col-0 pollen grains. (**A and B**) Scale bars, 5 µm. (**C**) Quantification of the number of turns (T) made by Col-0 pollen tubes on drug-treated and control papillae. Data are expressed as mean +/- s.e.m. Statistical difference was found between pollen tube path within DMSO (control) and oryzalin-treated papillae and was calculated using an adjusted Chi-Square test for homogeneity (2 degrees of freedom), p<0.01. N (DMSO)=12 stigmas, n(DMSO)=117 papillae, N (oryzalin)=16 stigmas, n(oryzalin)=149 papillae.

*Figure 4—source data 1* provides data for the quantification of the number of turns made by Col-0 pollen tubes on drug-treated and control papillae shown in C.

The online version of this article includes the following source data for figure 4:

**Source data 1.** Source data for the quantification of the number of turns made by Col-0 pollen tubes on drug-treated and control papillae shown in *Figure 4C*.

(*Figure 5—figure supplement 2B*). We then wondered whether alteration of the KATANIN function in pollen might affect pollen tube growth directionality on Col-0 or *ktn1-5* stigmas. Cross-pollination revealed that *ktn1-5* pollen tubes grew like Col-0 pollen tubes on stage-13 Col-0 papillae, while they coiled on *ktn1-5* papillae (*Figure 5—figure supplement 2C*). Seed counts showed that fertility slightly decreased when *ktn1-5* pollen was used to pollinate Col-0 stigmas, and was strongly affected when *ktn1-5* was used as female, irrespective of whether the pollen was Col-0 or *ktn1-5* (*Figure 5—figure supplement 2D*). This indicates that the fertilisation process is impaired in *ktn1-5* ovules. Altogether, these results show that the coiled phenotype observed on *ktn1-5* stigmas is only dependent on defects occurring at the papilla level, that the stigma receptivity of *ktn1-5* mutant is not altered and confirm that stigmatic CMTs, somehow, contribute to the directional growth of pollen tubes in papilla cells.

## Defects in papilla cell wall composition are not sufficient to affect Col-0 pollen tube behaviour

The *katanin1/fra2* mutant was initially described as a mutant impaired in cell wall biosynthesis and CMT array organisation (*Burk et al., 2001*). This prompted us to investigate whether other mutants affected in cell wall biogenesis might exhibit the coiled pollen tube phenotype. We selected mutants impaired in the cellulose synthase complex (*kor1.1*, *prc1* and *any1*), hemicellulose biosynthesis (*xxt1 xxt2*, *xyl1.4*) and pectin content (*qua2.1*), for which expression of the corresponding genes in stigma was confirmed (*Figure 5—source data 2*). Strikingly, none of the 6 cell wall mutants displayed the coiled pollen tube phenotype (*Figure 5—figure supplement 3A–F*). To ascertain that the cell wall composition was altered in the stigma papillae of *ktn1-5* and cell wall mutants, we undertook sugar analysis of *ktn1-5* and *xxt1 xxt2* stigmas and compared them with that of Col-0. We chose the *xxt1 xxt2* mutant because its cell wall composition had been already well described (*Cavalier et al., 2008*; *Xiao et al., 2016*). The relative abundance of monosaccharides in pistil cell walls varied between the three genotypes with major differences detected in the *ktn1-5* cell wall composition that had an increased abundance of glucose, this latter deriving either from hemicellulose and amorphous cellulose or from crystalline cellulose (*Figure 5—figure supplement 4A*). *ktn1-5* cell wall was characterized by the highest relative

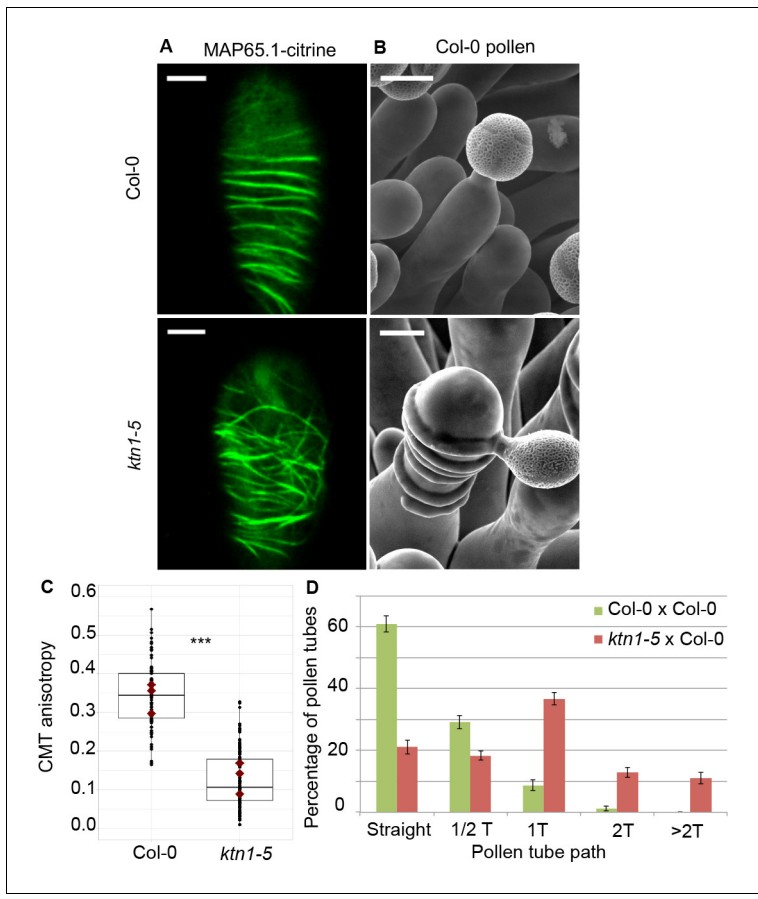

**Figure 5.** Effect of CMT organisation on pollen tube path. (**A**) Confocal images of papilla cells expressing MAP65.1-citrine in Col-0 and *ktn1-5* at stage 13. Scale bars, 5 μm. (**B**) SEM images of Col-0 and *ktn1-5* papillae pollinated with Col-0 pollen grains. Scale bar, 10 μm. (**C**) CMT anisotropy of Col-0 and *ktn1-5* papilla cells at stage 13. N(Col-0)=10 stigmas, n(Col-0)=106 papillae, N(*ktn1-5*)=11 stigmas, n(*ktn1-5*)=114 papillae. Statistical differences were calculated using a Shapiro-Wilk test to evaluate the normality and then a Wilcoxon test with ***p<0.01. (**D**) Quantification of the number of turns (T) made by Col-0 pollen tubes on *ktn1-5* and Col-0 papillae. Data are expressed as mean +/- s.e.m. Statistical difference was found between pollen tube path within *ktn1-5* and Col-0 papillae and was calculated using an adjusted Chi-Square test for homogeneity (2 degrees of freedom), p<0.01. N(Col-0)=27 stigmas, n(Col-0)=251 papillae, N(*ktn1-5*)=23 stigmas, n(*ktn1-5*)=327 papillae. *Figure 5—figure supplement 1* shows SEM images of the behaviour of Col-0 pollen tubes on Col-0 and *ktn1-5* papillae at stage 13. *Figure 5—figure supplement 2* illustrates the receptivity and fertility of the *ktn1-5* mutant. *Figure 5—figure supplement 3* shows that mutants impaired in cell wall genes behave like Col-0. *Figure 5—figure supplement 4* gives the saccharide composition of Col-0, *ktn1-5* and *xxt1 xxt2* stigmatic cell walls. *Figure 5—figure supplement 5* compares the length and width of papilla cells from Col-0, *ktn1-5*, *xxt1 xxt2* and *any1*. *Figure 5—source data 1* provides source data for the quantification of CMT anisotropy of Col-0 and *ktn1-5* papilla cells at stage 13 shown in **C**, for the quantification of the number of turns made by Col-0 pollen tubes on *ktn1-5* and Col-0 papillae shown in **D**, and source data for the *Figure 5—figure supplement 2*, *Figure 5—figure supplement 3* and *Figure 5—figure supplement 4*. *Figure 5—source data 2* provides information on the six cell wall genes analysed and shows that they are expressed in stigmas.

The online version of this article includes the following source data and figure supplement(s) for figure 5:

**Source data 1.** Source data for the quantification of CMT anisotropy of Col-0 and *ktn1-5* papilla cells at stage 13 shown in *Figure 5C* and the quantification of the number of turns made by Col-0 pollen tubes on *ktn1-5* and Col-0 papillae shown in *Figure 5D*.

**Source data 2.** Table of cell wall mutants analysed.

**Figure supplement 1.** Col-0 pollen tube behaviour on Col-0 and *ktn1-5* papillae at stage 13.

**Figure supplement 2.** *ktn1-5* is impaired in female receptivity and fertility.

**Figure supplement 3.** Quantification of the number of coils made by Col-0 pollen tubes on papillae from cell-wall mutants at stage 13.

*Figure 5 continued on next page*

*Figure 5 continued*

**Figure supplement 4.** Cell wall composition is altered in *katanin1* and different from *xxt1 xxt2*.
**Figure supplement 5.** Size of the papilla cells of Col-0 and cell wall mutants.

abundance of crystalline cellulose, which was 25% higher than in Col-0 and *xxt1 xxt2*. To explore whether side chains of hemicellulose might also be affected in the *katanin* mutant, we performed an enzymatic fingerprinting of xyloglucans from dissected stigmas of Col-0 and *ktn1-5*; *xxt1xxt2* was not analysed as it lacks xyloglucan side chains. We found in *ktn1-5* a decrease in relative abundance of short side chains with terminal xylose (XXG and XXXG) and an increase in the longer ones with terminal fucose (e.g XXFG) (*Figure 5—figure supplement 4B*). The chemical analysis confirms that the cell wall composition is altered in *ktn1-5* stigmas and presents some specific features compared with Col-0 and *xxt1 xxt2* stigmas. However, combined with the results of pollination on the six cell wall mutants, we cannot establish a causal effect of chemical alterations on the pollen tube coiled phenotype. This suggests that the relation between CMT organisation, cell wall composition and pollen tube trajectory is more complex than anticipated.

## Shape of papilla cells is not a determining factor of the pollen tube phenotype

Because CMTs guide cellulose deposition, they also control the directional elongation of plant cells (*Baskin, 2005*). Indeed, *katanin1/fra2* cells are wider and shorter than wild type (*Bichet et al., 2001*; *Burk et al., 2001*), and we may predict that the shape of papilla cells in the mutant might be similarly altered. The contribution of stigmatic CMTs to pollen tube growth may thus be mediated by papilla cell shape only; for instance, wider papilla cells in *ktn1-5* would promote pollen tube coiling. To check that possibility, we measured the length and width of papilla cells in Col-0 (at stage 13 and 15), *ktn1-5, xxt1 xxt2* and *any1* (*Figure 5—figure supplement 5*). We did not find any correlation between papilla length and the coiled phenotype. Indeed, coiled phenotype was observed in stage-15 WT papillae that were longer than those at stage-13, as well as in *ktn1-5* papillae that had a length similar to stage-13 WT papillae. The correlation between papilla width and coiled phenotype was also not clear-cut. As anticipated, *ktn1-5* mutant exhibited wider papillae than Col-0. However, *xxt1 xxt2* and *any1* papillae were also wider than the WT but did not display the coiled phenotype. Altogether, this suggests that papilla morphology is not sufficient to explain the coiled phenotype.

## Mechanical properties of the cell wall are disturbed in *ktn1-5* papilla cells

Because the main role of CMTs in plant cells is to guide the trajectory of CSCs, thereby impacting the mechanical anisotropy of the cell wall, we analysed the cell walls of Col-0 and *ktn1-5* papillae following pollination. First, using Transmission Electron Microscopy (TEM), we found Col-0 pollen tubes penetrated the cuticle and grew between the two layers of the papilla cell wall, as previously described (*Kandasamy et al., 1994*), in both Col-0 and *ktn1-5* papilla cells (*Figure 6A*). We did not detect any significant difference in the ultrastructure of cell walls (*Figure 6—figure supplement 1*). Interestingly, as the pollen tube progresses through the papilla cell wall, it generates a bump (external deformation) and an invagination (internal deformation) in the cell wall (*Figure 6A*). As such deformations could reflect differences in wall properties, we quantified the external (extD) and internal (intD) deformation following pollination of Col-0 and *ktn1-5* stigmas. To visualize more clearly this deformation, we pollinated stigmas expressing the plasma membrane protein LTI6B fused to GFP (LTI6B-GFP) with pollen whose tube was labelled with the red fluorescent protein RFP driven by the ACT11 promoter (*Rotman et al., 2005*; *Figure 6B*). We found that Col-0 pollen tubes grew with almost equal extD and intD values in Col-0 cell wall. However, the ratio between extD and intD was about three when Col-0 pollen tubes grew in *ktn1-5* papilla cells (*Figure 6B–D*). These quantitative data were consistent with the observation that pollen tubes appeared more prominent on *ktn1-5* stigmatic cells using SEM (*Figure 5B*; *Figure 5—figure supplement 1B*). Assuming that stigmatic cells are pressurized by their turgor pressure, this may reflect the presence of softer walls in *ktn1-5* papillae. To test this hypothesis, we assessed the stiffness of Col-0 and *ktn1-5* papilla cell walls using Atomic Force Microscopy (AFM) with a 200 nm indentation. We found that cell wall stiffness in *ktn1-*

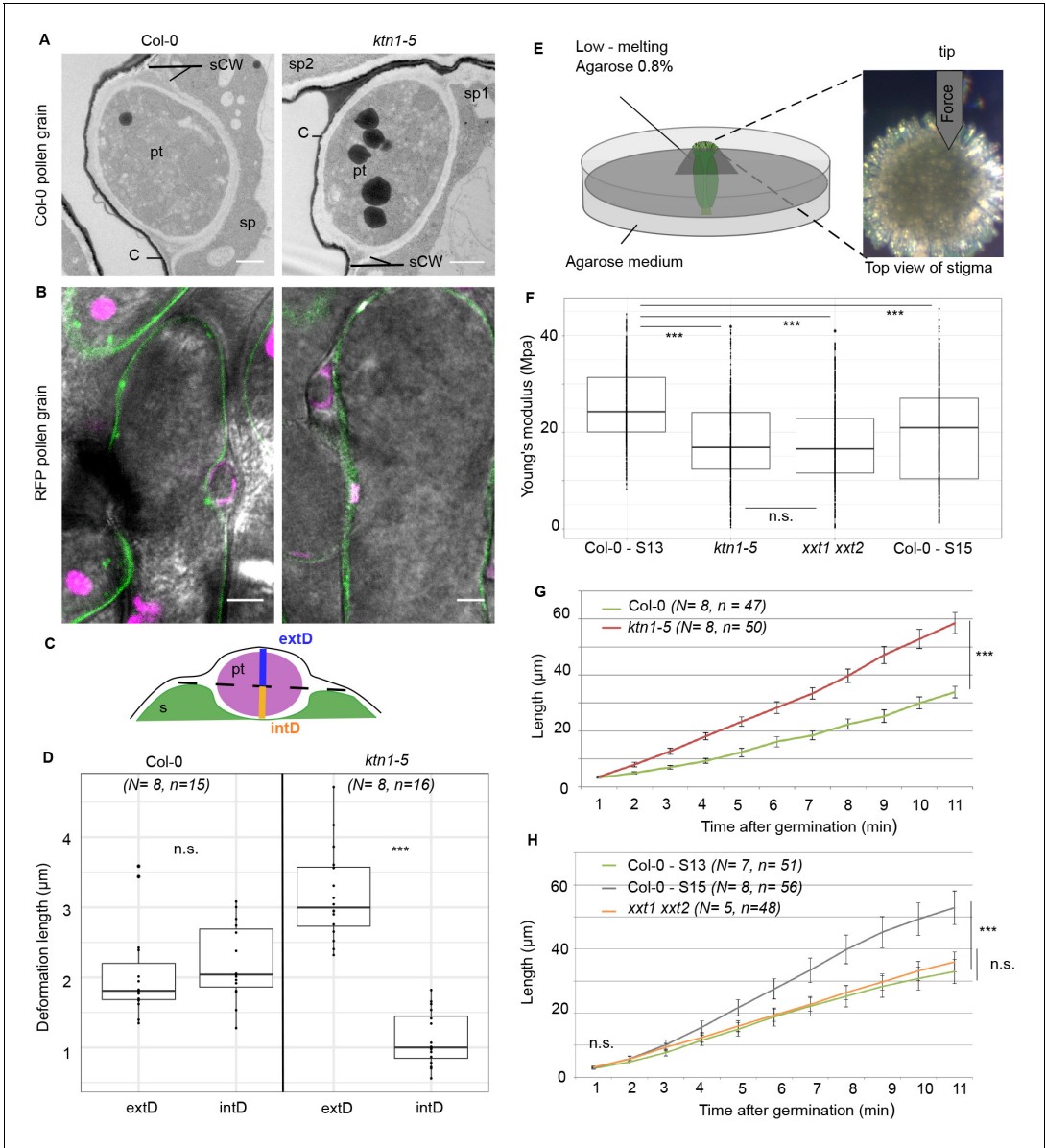

**Figure 6.** Mechanical properties of papilla cell walls. (**A**) Location of a Col-0 pollen tube in the cell-wall of Col-0 and *ktn1-5* papillae by TEM. pt = pollen tube, C = stigma cuticle, sCW = stigma cell wall, sp = stigma papilla. Scale bar, 1 μm. (**B**) Confocal images of Col-0 and *ktn1-5* papillae expressing the plasma membrane marker LTI6B-GFP pollinated with an RFP-expressing pollen. Scale bar, 5 μm. (**C**) Diagram showing the procedure used for evaluating the external (extD) and internal (intD) deformations made by Col-0 pollen tubes. (**D**) External and internal deformations caused by Col-0 pollen tube growth in Col-0 and *ktn1-5* papillae. (**E**) Drawing of the AFM experimental setup. Dissected pistils were inserted in agarose medium and fixed with low-melting agarose for measurements. (**F**) Young's modulus values of the papilla cell wall for Col-0 at stage 13 (N = 4 stigmas, n = 8 papillae), *ktn1-5* (N = 5 stigmas, n = 9 papillae), *xxt1 xxt2* (N = 4 stigmas, n = 11 papillae) and Col-0 at stage15 (N = 4 stigmas, n = 10 papillae. (**G**) Mean of travel distances made by Col-0 pollen tubes in Col-0 and *ktn1-5* papillae. (**H**) Mean of travel distances made by Col-0 pollen tubes in papillae of Col-0 at stage 13, Col-0 at stage 15, *xxt1 xxt2* at stage 13. D,F and G,H: Statistical differences were calculated using a Shapiro-Wilk test to evaluate the normality and then a T-test. ***p<0.01, n.s. = not significant. For H, we found a significant difference (***p<0.01) between Col-0 at stage 13 and 15, but not significant (n.s.) between Col-0 at stage 13 and *xxt1 xxt2* at stage 13. *Figure 6—figure supplement 1* shows the similar ultrastructural features of Col-0 and *ktn1-5* papilla cell walls. *Figure 6—source data 1* are values measured to define the mechanical properties of the papilla cell walls shown in **D, F, G, H**.

The online version of this article includes the following source data and figure supplement(s) for figure 6:

**Source data 1.** Source data for defining the mechanical properties of the papilla cell walls shown in *Figure 6D*, *Figure 6F*, *Figure 6G*, *Figure 6H*.
**Figure supplement 1.** Ultrastructural features of Col-0 and *ktn1-5* papilla cells.

5 papilla cells was about 30% lower than that in stage 13 WT cells (*Figure 6E,F*). To confirm this result, we investigated the stiffness of the papilla cell wall on WT stigmas at stage 15, where increased coiled pollen tubes were detected (*Figure 2A,B*). We found the cell wall to be softer than that of papillae at stage 13 but stiffer than that of the *ktn1-5* (*Figure 6E,F*). We then reasoned that the presence of softer walls should also affect the pollen tube growth rate, stiffer walls reducing growth. Thus, measuring the growth rate would provide an indication of the resistance force encountered by the tube while growing in the papilla cell wall. We thus monitored the growth rate of Col-0 pollen tubes in Col-0 and *ktn1-5* papillae. We found that pollen tubes grew faster (~x 1.8) within *ktn1-5* papillae (*Figure 6G*). Similarly, we found that pollen tube growth on stage 15 papillae was faster (~x 1.6) than on stage 13 papillae (*Figure 6H*). These results suggest that the coiled phenotype is associated with a higher pollen tube growth rate and hence, that papilla cell walls of *ktn1-5* and stage-15 papillae exhibit less resistance to tube penetration. Altogether, these data suggest that KATANIN-dependent mechanical properties of papilla cell wall play a role in pollen tube guidance.

## Mechanical anisotropy of the papilla cell wall mediates pollen tube directionality

Because CMT disorganisation in papillae affects both wall stiffness and mechanical anisotropy, we next investigated the relative contribution of these two parameters in pollen tube growth. Hypocotyl cells of the *xxt1 xxt2* double mutant were reported to display CMT orientation defects and reduced stiffness of the cell wall (*Xiao et al., 2016*), features that we found in *ktn1-5* papilla cells. However, growing pollen tubes on *xxt1 xxt2* stigmas did not coil and grew like on WT stigmas, questioning whether the cell wall stiffness of stigmatic cells was actually affected in the mutant. Using AFM, we found the cell wall of *xxt1 xxt2* papillae to be about 30% softer than that of Col-0 papillae, i.e. very similar to that of *ktn1-5* (*Figure 6F*). Despite this similarity, the pollen tube growth rate in *xxt1 xxt2* stigmas was identical to Col-0 stigmas at stage 13 (*Figure 6H*). In addition, contrary to *ktn1-5*, pollen tubes were not prominent while growing in *xxt1 xxt2* papilla cell walls (*Figure 5—figure supplement 3G*). These results suggested that although *ktn1-5* and *xxt1 xxt2* papillae had similar cell wall stiffness as inferred from AFM measurements, they exhibited different mechanical constraints to pollen tube growth. Because of the close relationship between CMT organisation, CMF deposition and cell wall rigidity (*Xiao et al., 2016*; *Xiao and Anderson, 2016*), we suspected that CMT and CMF organisation might be different between the two mutants and be the possible causal agent of the mechanical differences and related tube coiled phenotype observed in *ktn1-5*. To check this possibility, and to gain insight into the predominant cellulose pattern in the stigmatic cell walls, we stained CMFs by using Direct Red 23. We found cellulose fibres to be highly aligned and slanted to the longitudinal axis of the papilla in stage 13 Col-0 stigmas, whereas *ktn1-5* and stage 15 Col-0 papillae both displayed a clearly disordered CMF pattern (*Figure 7A*) with CMFs forming thicker and more spaced bundles (*Figure 7B*). By contrast, CMFs in *xxt1 xxt2* papillae at stage 13 organised in a dense and well oriented pattern of fibres, mostly perpendicular to the papilla long axis, which resembled Col-0 stage 13 CMF organisation (*Figure 7A,B*). These results suggest that KTN1 loss-of-function alters mechanical properties of the papilla cell wall in a complex manner, lowering both its stiffness to pressure applied perpendicularly to its surface (e.g., by the AFM indenter) and its resistance to pollen tube progression. This latter modification appears as the main cause of the pollen tube coiled phenotype. By contrast, loss of XXT1 and XXT2 functions, by perturbing cell wall stiffness to indenter pressure only, does not induce change in pollen tube growth. Altogether, our data suggest that the mechanical anisotropy of papilla cell walls plays a key role in pollen tube trajectory.

## Discussion

Previous work reported that rearrangements of actin microfilaments (*Iwano et al., 2007*; *Rozier et al., 2020*) and destabilization of CMTs (*Samuel et al., 2011*) occur in stigmas following compatible pollination in Brassicaceae species. However, CMT pattern and dynamics during papilla development have never been described. Our data show that during the course of stigma maturation, which is associated with papilla cell elongation, CMT bundles progressively move from perpendicular (anisotropic) to the elongation axis at stage 12 to disorganised (isotropic) at stage 15. This correlates with the known CMT dynamics during elongation in plant cells, where CMT arrays are

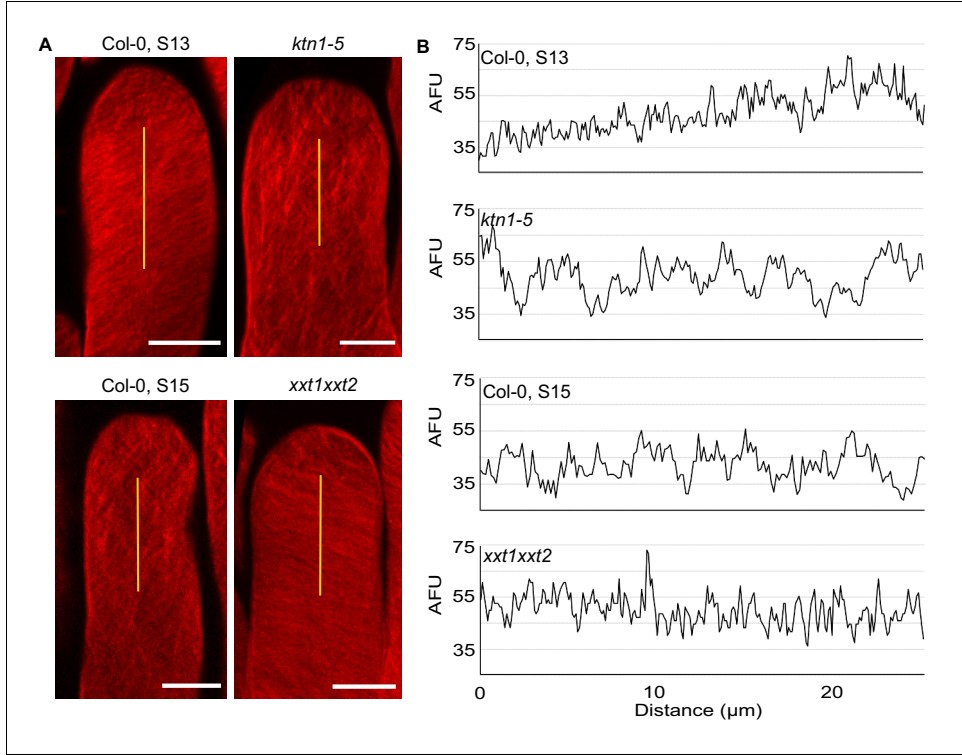

**Figure 7.** Cellulose microfibril organisation in Col-0 stage 13, *ktn1-5*, Col-0 stage 15 and *xxt1 xxt2* papillae. (**A**) 3D-projections of z-stack images of Col-0 stage 13, *ktn1-5*, Col-0 stage 15 and *xxt1 xxt2* papilla cells stained with Direct Red 23 dye. Scale bars, 10 μm. (**B**) Plot profiles of the fluorescence intensity (AFU, Arbitrary Fluorescence Units) along the yellow lines. Note the similarities of profiles between Col-0 stage 13 and *xxt1 xxt2* on one hand, and between *ktn1-5* and Col-0 on the other hand. At least eight papillae per genotype were observed.

highly anisotropic in young cells and become more isotropic as cells differentiate (*Baskin, 2005*; *Landrein and Hamant, 2013*).

During invasion of plant cells by fungal or Oomycete pathogens or upon mechanical stimulation, the network of microtubules is rearranged, generally associated first with the depolymerization of CMTs at the contact site and later with the formation of an array of microtubules surrounding the invading organism or microneedle (*Hardham, 2013*). Interestingly, penetration of the pollen tube in the papilla cell wall does not lead to such microtubule reorganisation (*Figure 3*). This suggests that at the cellular level, the invading pollen tube does not elicit a defence-like response in the papilla, which accepts and supports pollen tube growth. In addition, we show that when the stigma is manually pressed between slide and coverslip, the mechanical stimulation applied perpendicularly to the wall leads to CMT fragmentation (*Figure 3—figure supplement 1*). This indicates that although the papilla microtubule network is sensitive to mechanical forces, the pressure exerted by the pollen tube is not strong enough to perturb CMT organisation. One possible explanation is that the penetration force developed by the growing pollen tube acts mainly in plane to the cell wall surface and hence would only have a little effect on the papilla cell, contrary to a pressure applied perpendicularly to the wall. Our data are in contradiction with the earlier observation that growing pollen tube provokes fragmentation of CMTs in the papilla (*Samuel et al., 2011*). This discrepancy is likely to relate to the microscopic approaches used between the two studies, live imaging and a fluorescent microtubule probe in our case, chemically fixed material and immunostaining of microtubules in *Samuel et al., 2011*.

The progressive randomisation of CMT orientation we observed in papilla cells during ageing is accompanied by an increased coiled growth of pollen tubes in papillae. Similarly, when CMTs are destabilized by the microtubule depolymerizing drug oryzalin, coiled pollen tubes are more frequently observed compared with untreated control stigmas. These results reveal a link between the stigmatic CMT cytoskeleton organisation and the trajectory that pollen tube takes while growing in

the papilla cell wall, although the effects of oryzalin on cell wall properties remain unknown. The fact that the most striking effect on pollen tube growth was found on *ktn1-5* mutant suggests that the coiled phenotype depends not only on the CMT organisation but implicates other factors. Among various phenotypic alterations described in loss-of-function mutants for *KTN1*, are the impaired cell mechanical properties and cell elongation, defects in CMT organisation, cell wall composition and CMF orientation (*Burk et al., 2001*; *Burk and Ye, 2002*; *Ryden et al., 2003*). Interestingly, we found that the protuberance of pollen tubes at the surface of *ktn1-5* papillae was associated with a faster growth rate and a lower rigidity of the cell wall compared with Col-0 (*Figure 6*). This places mechanics of the cell wall as a likely component involved in the coiled phenotype. Surprisingly, of the six cell wall mutants we analysed, including *xxt1 xxt2* and *prc1* known to exhibit both abnormal CMT organisation and softer cell walls, none induced the coiled phenotype displayed by *ktn1-5* papillae. More remarkably, despite the similar stiffness of *xxt1 xxt2* and *ktn1-5* papilla cell walls as measured by AFM, pollen tubes behaved differently on these two stigma types, pollen tubes not only coiling but also growing faster on *ktn1-5* stigmas. This observation is in accordance with the recent findings that changes in indentation properties of plant cell walls do not correlate with changes in tensile stiffness as described in cell-free strips of onion epidermal walls (*Zhang et al., 2019*). In our experiments, indentations tell us about the elasticity of the wall to a pressure normal to the plane of the wall surface (i.e. out-of-plane mechanics), while pollen tube growth rates inform about the resistance of the wall to a force in-plane to the papilla wall (i.e. in-plane mechanics). Several lines of evidence lead us to suggest that mechanical properties of the cell wall are not identical in the two mutants. Previous work showed that cellulose crystallinity is one factor implicated in cell wall mechanics. Indeed, the epidermal cells of the inflorescence stem in the *any1* mutant have the relative amount of crystalline cellulose significantly reduced, while the total content of cellulose in the cell wall is unaltered, and this is accompanied by reduced stiffness of the cell wall (*Fujita et al., 2013*; *Altartouri et al., 2019*). Our chemical analysis reveals that the cell wall composition is different between *xxt1 xxt2* and *ktn1-5* stigmas, with this latter having 25% more crystalline cellulose, and hence should exhibit stiffer cell walls (*Figure 5—figure supplement 4*). This data is at odds with AFM measurements but a possible explanation would be that the two layers detected by TEM in the papilla cell wall (*Figure 6—figure supplement 1*) display different mechanical properties, the inner layer being stiffer than the outer one in the *ktn1-5* mutant. This may explain the protruding growth of pollen tubes observed in *ktn1-5* papillae, the tube pushing away the softer outer layer but not the inner (*Figure 6A–D*). Interestingly, *Zhang et al., 2019* reported that cellulose networks largely determine in-plane mechanics whereas out-of-plane mechanics depends on both homogalacturonan (pectin) and cellulose networks. Cell walls of the *qua2-1* mutant are specifically affected in pectins, with a 50% reduction in homogalacturonan content compared with Col-0 without observable changes in other polysaccharides (*Mouille et al., 2007*). In light of these data, it is noteworthy that papilla cells of the *qua2-1* mutant do not induce pollen tube turns (*Figure 5—figure supplement 3*), which suggests that pectins play no significant role in the pollen tube phenotype and that the main factor contributing to pollen tube guidance is in-plane mechanics, and hence relies on the cellulose network. The orientation of the rigid CMFs is another determinant of cell wall mechanics. The *ktn1-5* mutant was described to have a severe reduction in cell length and an increase in cell width in all organs (*Burk et al., 2001*). This cellular phenotype was attributed to the distorted deposition of CMFs correlated with the isotropic orientation of CMTs, whereas in WT cells, CMFs like CMTs are oriented perpendicularly to the elongation axis (*Burk and Ye, 2002*). Similarly, we found that orientation of CMFs is also altered in *ktn1-5* papillae (*Figure 7*) and this was associated with larger papilla cells (*Figure 5—figure supplement 5*). Hence, we may assume that mechanical anisotropy, described as the cell wall anisotropy made by the orientation of the rigid CMFs (*Sassi et al., 2014*), is impaired in *ktn1-5* papillae. Contrary to the *ktn1-5* mutant, papillae of the *xxt1 xxt2* double mutant display an anisotropic CMF pattern with parallel fibres approximatively oriented transversally to the papilla axis. These data are in agreement with those reported for etiolated hypocotyl cells, where CMFs are largely parallel to one another, straighter than in WT and oriented almost transversely to the long axis of the cell (*Xiao et al., 2016*). In addition, we found the papilla cell shape of *xxt1 xxt2* stigmas to be closer to that of Col-0 than that of *ktn1-5* papillae, which is consistent with an anisotropic growth of the papillae in the *xxt1 xxt2* mutant compared with *ktn1-5*. Apart from the similar stiffness of papilla cell walls as deduced from AFM measurements, the main differences between *ktn1-5* and *xxt1 xxt2* cell walls are the cell wall composition, orientation of CMFs, and possible changes in molecular connections between cell wall

components and plasma membrane and/or cytoskeleton proteins. Indeed, a recent proteomic study revealed that loss of KATANIN function is associated with the decrease in abundance of several cytoskeleton proteins, such as profilin 1, actin-depolymerizing factor 3 and actin 7 (*Takáč et al., 2017*), whereas targeted quantitative RT-PCR showed that several Microtubule-associated protein (MAP) and wall signal receptor genes showed lower expression levels in *xxt1 xxt2* (*Xiao et al., 2016*). In this latter study, *KTN1* expression level was shown to be unchanged compared with Col-0. Altogether, our study suggests that KTN1, by maintaining the papilla mechanical anisotropy, has a key function in mediating early pollen tube guidance on stigma papillae.

The coiled phenotype was not only observed in *ktn1-5* but also in the WT Col-0 papillae at stage 15. Remarkably, several lines of evidence reveal that papilla cells from stage 15 share common features with *ktn1-5* papillae, such as CMT and CMF increased isotropy, decreased stiffness of the cell wall as deduced from AFM and less resistance of the wall to pollen tube growth. Isotropic orientation of the cytoskeleton at stage 15 is likely to relate to cell elongation, which is known to be accompanied by cytoskeleton reorganisation (*Crowell et al., 2011*; *Zhang et al., 2014*). At the organ level, it has been suggested that the mechanical anisotropy of the wall restrains organ emergence (*Sassi et al., 2014*). The authors propose that for the same wall stiffness, a cell wall with isotropic properties would lead to larger outgrowth than a wall with anisotropic properties. Our data are consistent with this hypothesis, albeit at the subcellular scale, since large protuberance of papilla wall following pollen tube growth is observed in *ktn1-5* papilla cells, exhibiting walls with isotropic properties.

It remains unclear how mechanical anisotropy guides pollen tube growth. We can hypothesise that as pollen tube grows inside the wall, it encounters recently deposited CMFs on the inner side of the wall (facing the cytoplasm) and older CMFs on the outer side. It is likely that these layers have different mechanical properties related to CMF orientation (*Baskin, 2005*). KATANIN, by acting on CMT dynamics and CMF organisation, would fine-tune the mechanical properties of the matrix through which the pollen tube grows. It is worth noting that growth rate of pollen tubes germinated in vitro is slowed down when pollen tubes pass through a microgap of a microfluidic device, the tubes adapting their invasive force to the mechanical constraints (*Sanati Nezhad et al., 2013*). Based on measurements of pollen tube growth rates, our data show that the tube tip, while progressing in the papilla wall, senses the mechanical features of its environment and reacts accordingly. Hence, it reveals some unanticipated internal and hidden properties of the cell wall.

Our study shows that mechanics plays a key role in early pollen tube guidance in the papilla cell. This role is mediated by a specific CMT/CMF organisation and mechanical anisotropy of the papilla cell, which both are dependent on KTN1. Importantly, these specific mechanical properties of the stigmatic cells prevent emerging pollen tubes to grow upward on papillae and straighten pollen tube direction, helping the tube to find its correct path to the stylar transmitting tract. Our findings also raise the question about the biological importance of the mechanical changes that occur during stigma development. Indeed, for aged stigmas, modifications of the mechanical properties of the cell wall, accompanied by an acceleration of pollen tube growth, could be seen as beneficial for the plant by favouring ultimate fertilization and seed set on old flowers, and hence supporting dissemination of the species. This assumption is particularly relevant given that the pistil length increases with ageing and similarly the journey the pollen tube has to travel to reach the ovules. A recent work showed that ageing is associated with decreased fertility due to stigma senescence, which is initiated at stage 16–17 (*Gao et al., 2018*). The faster the pollen tube will grow in old stigmas, the more chance it will have to circumvent papilla cell death and hence to fertilize the ovule. We may suggest that KTN1, by mediating mechanical anisotropy of stigmatic cells and promoting pollen tube growth in old stigmas, has played an evolutionary role in the success of fertilization.

To conclude, we uncovered a yet unexpected function for KTN1 in early pollen tube guidance on the stigma. In addition, our study also clearly unveils that the mechanical properties of one single cell (e.g., the stigmatic papilla) impact the behaviour of its neighbouring cell (e.g., the pollen tube).

## Materials and methods

**Key resources table**

*Continued on next page*

*Continued*

| Reagent type (species) or resource | Designation | Source or reference | Identifiers | Additional information |
|---|---|---|---|---|
| Reagent type (species) or resource | Designation | Source or reference | Identifiers | Additional information |
| Genetic reagent *Arabidopsis thaliana* | ktn1-5 | *Lin et al., 2013* | AT1G80350 | |
| Genetic reagent *Arabidopsis thaliana* | xxt1 xxt2 | *Cavalier et al., 2008* | AT3G62720 AT4G02500 | |
| Genetic reagent *Arabidopsis thaliana* | prc1.1 | *Fagard et al., 2000* | AT5G64740 | |
| Genetic reagent *Arabidopsis thaliana* | qua2.1 | *Mouille et al., 2007* | AT1G78240 | |
| Genetic reagent *Arabidopsis thaliana* | xyl1.4 | *Sampedro et al., 2010* | AT1G68560 | |
| Genetic reagent *Arabidopsis thaliana* | kor1.1 | *Nicol et al., 1998* | AT5G49720 | |
| Genetic reagent *Arabidopsis thaliana* | any1 | *Fujita et al., 2013* | AT4G32410 | |
| Genetic reagent *Arabidopsis thaliana* | pSLR1::MAP65.1-citrine | This paper | | Transgenic line expressing MAP65.1-citrine in stigmas, request to RDP laboratory, ENS Lyon, France |
| Genetic reagent *Arabidopsis thaliana* | pSLR1::LTI6B-GFP | This paper | | Transgenic line expressing LTI6B-GFP in stigmas used in *Figure 6*, request to RDP laboratory, ENS Lyon, France |
| Genetic reagent *Arabidopsis thaliana* | pACT11::RFP | This paper | | Transgenic line expressing RFP under the ACTIN11 promoter used to visualize pollen and pollen tubes in *Figure 3* and *Figure 6*, request to RDP laboratory, ENS Lyon, France |
| Chemical compound, drug | Oryzalin | Chemical Service, Supelco | 36182 | 833 µg/mL |
| Chemical compound, drug | RedDirect23 | Sigma-Aldrich | 212490 | 0.02% (w/v) |

*Continued on next page*

*Continued*

| Reagent type (species) or resource | Designation | Source or reference | Identifiers | Additional information |
|---|---|---|---|---|
| Chemical compound, drug | Adigor | Syngenta | | 2.5% (v/v in water) |
| Software, algorithm | ImageJ | | RRID:SCR_003070 | https://imagej.net |
| Software, algorithm | FibrilTool | *Boudaoud et al., 2014* | RRID:SCR_016773 | https://biii.eu/fibriltool |
| Software, algorithm | Rstudio | *RStudio Team, 2015* | RRID:SCR_000432 | https://rstudio.com/ |

## Plant materials and growth conditions

*Arabidopsis thaliana*, ecotype Columbia (Col-0), *Arabidopsis* transgenic plants generated in this study and *Arabidopsis* mutants were grown in soil under long-day conditions (16 hr of light/8 hr of dark, 21 °C / 19°C) with a relative humidity around 60%. *ktn1-5* (SAIL_343_D12), *xxt1xxt2*, *prc1.1*, *qua2.1*, *xyl1.4*, *kor1.1* and *any1* mutant lines were described previously (*Cavalier et al., 2008*; *Fagard et al., 2000*; *Fujita et al., 2013*; *Lin et al., 2013*; *Mouille et al., 2007*; *Nicol et al., 1998*; *Sampedro et al., 2010*; *Shoji et al., 2004*). All mutants were in Col-0 background except *kor1.1* which was in WS. All stigmas were analysed at stage 13 of flower development (*Smyth et al., 1990*) except when specified.

## Plasmid construction

We used the Gateway technology (Life Technologies, USA) and two sets of Gateway-compatible binary T-DNA destination vectors (*Hellens et al., 2000*; *Karimi et al., 2002*) for expression of transgenes in *A. thaliana*. The DNA fragment containing the *Brassica oleracea SLR1* promoter was inserted into the pDONP4-P1R vector. The 165 bp-*LTI6B* fragment was introduced into the pDONR207 vector. *MAP65* gene spanning the coding region from start to stop codons was introduced into the pDONR221 vector. CDS from citrine or GFP were cloned into the pDONP2R-P3 vector. Final constructs, pSLR1::MAP65-citrine and pSLR1::LTI6B-GFP were obtained by a three-fragment recombination system (Life Technologies) using the pK7m34GW and the pB7m34GW destination vectors, respectively. We generated a pACT11::RFP construct by amplifying the promoter of the *A. thaliana ACTIN-11* gene and cloning it into the pGreenII gateway vector in front of the RFP coding sequence.

## Generation of transgenic lines and crossing

Transgenic lines were generated by *Agrobacterium tumefaciens*-mediated transformation of *A. thaliana* Col-0 as described (*Logemann et al., 2006*). Unique insertion lines, homozygous for the transgene were selected. We introduced the pSLR1::LTI6B-GFP or pSLR1::MAP65-citrine construct in *ktn1-5* background by crossing and further selecting the progeny on antibiotic containing medium.

## Microscopy
### Confocal microscopy

Flowers at stages 12 to 15 (*Smyth et al., 1990*) collected from fluorescent lines were emasculated and stigmas were observed under a Zeiss LSM800 microscope (AxioObserver Z1) using a 40x Plan-Apochromat objective (numerical aperture 1.3, oil immersion). For pollination experiments using the MAP65.1-citrine line, flowers were emasculated and pollinated with mature pollen from the pACT11::RFP line. 30 min after pollination, stigmas were observed under the confocal microscope. Citrine was excited at 515 nm and fluorescence detected between 530 and 560 nm. GFP was excited at 488 nm and fluorescence detected between 500 and 550 nm. RFP was excited at 561 nm and fluorescence detected between 600 and 650 nm.

### Live imaging

Flowers at stage 13 and 15 were emasculated and pollinated on plants with mature pollen from the pACT11::RFP line. Immediately after pollination, stigmas were mounted between two coverslips maintained separated by four grease plugs placed at each coverslip corner. To maintain a constant humidity without adding liquid directly on the stigma surface, we used a wet piece of tissue in contact with the base of the stigma. Pollinated stigmas were observed under a Zeiss microscope (AxioObserver Z1) equipped with a spinning disk module (CSU-W1-T3, Yokogawa) using a 40x Plan-Apochromat objective (numerical aperture 1.1, water immersion). Serial confocal images were acquired in the entire volume of the stigma every 1 μm and every minute. Images were processed with Image J software and pollen tube lengths were measured.

### Atomic Force Microscopy

Pistils were placed straight in a 2% agar MS medium and 0.8% low-melting agarose was added up to a certain level where the papilla cells were maintained well immobilised in agarose while leaving the top accessible to the indenter. This set up allowed accurate measurements of cell wall stiffness on the dome-shaped top of papilla cells. AFM indentation experiments were carried out with a Catalyst Bioscope (Bruker Nano Surface, Santa Barbara, CA, USA) that was mounted on an optical microscope (MacroFluo, Leica, Germany) equipped with a x10 objective. All quantitative measurements were performed using standard pyramidal tips (RFESP-190 (Bruker)). The tip radius given by the manufacturer was 8–12 nm. The spring constant of the cantilever was measured using the thermal tune method and was 35 N/m. The deflection sensitivity of the cantilever was calibrated against a sapphire wafer. All experiments were made in ambient air at room temperature. Matrix of $10 \times 10$ measurements (step 500 nm) was obtained for each papilla, with a 1μN force. The Young's Modulus was estimated using the Nanoscope Analysis (Bruker) software, using the Sneddon model with a < 200 nm indentation.

### Environmental Scanning Electron Microscopy (SEM)

Flowers from stages 12 to 15 were emasculated and pollinated on plants with mature WT pollen. One hour after pollination, pistils were cut in the middle of the ovary, deposited on a SEM platform and observed under Hirox SEM SH-3000 at −20°C, with an accelerating voltage of 15kV. Images were processed with ImageJ software and pollen tube direction was quantified by counting the number of turns made by the tube, only on papillae that received one unique pollen grain.

### Transmission Electron Microscopy

Stage 13 flowers were emasculated and pollinated on plants with mature WT pollen. One hour after pollination, pistils were immersed in fixative solution containing 2.5% glutaraldehyde and 2.5% paraformaldehyde in 0.1 M phosphate buffer (pH 7.2) and after 4 rounds of 30 min vacuum, they were incubated in fixative for 12 hr at room temperature. Pistils were then washed in phosphate buffer and further fixed in 1% osmium tetroxide in 0.1 M phosphate buffer (pH 7.2) for 1.5 hr at room temperature. After rinsing in phosphate buffer and distilled water, samples were dehydrated through an ethanol series, impregnated in increasing concentrations of SPURR resin over a period of 3 days before being polymerized at 70°C for 18 hr, sectioned (65 nm sections) and imaged at 80 kV using an FEI TEM tecnaiSpirit with 4 k x 4 k eagle CCD.

### Anisotropy estimation

Flowers from MAP65 lines were emasculated and stigmas were observed under confocal microscope. Images were processed with ImageJ software and quantitative analyses of the average orientation and anisotropy of CMTs were performed using FibrilTool, an ImageJ plug-in (*Boudaoud et al., 2014*). Anisotropy values range from 0 to 1; 0 indicates pure isotropy, and one pure anisotropy.

### Cellulose staining

A droplet of Adigor was applied at 2.5% (v/v in water) on the stigma during 4 hr to facilitate subsequent chemical treatments. As described for hypocotyls (*Landrein and Hamant, 2013*), pistils were incubated in 12.5% glacial acetic acid for 1 hr and dehydrated in 100% and then 50% ethanol for 20

min each. Pistils were then washed in water during 20 min and stored in 1M KOH during 2 days. Pistils were then stained using 0.02% (w/v) Direct Red 23 dye (Sigma-Aldrich) during 4 hr and washed with distilled water. Pistils were observed with a mRFP filter (excited at 561 nm) under a Zeiss LSM 880 confocal microscope using a 63x Plan-Apochromat objective (numerical aperture 1.4, oil immersion). Stigmas were imaged by taking a z-stack of 0.2 µm sections in the papilla cells. Image analysis was performed by doing a 3D-projection and measuring the fluorescence intensity longitudinally along the papilla axis using ImageJ software.

## Membrane deformation estimation

Flowers from LTI6B lines were emasculated and pollinated with mature pollen from the pACT11:: RFP line. 20 min after pollination, stigmas were observed under confocal microscope. Serial confocal images every 1 µm encompassing the entire volume of the stigma were recorded and processed with ImageJ software. Plasma membrane deformation was estimated by choosing the slide from the stack that corresponded to the focus plan of the contact site with the RFP-labelled pollen tube. On the bright field image corresponding to the selected slide, a line was drawn connecting the two ridges of the invagination of the papilla. Two perpendicular lines, one toward the exterior (ExtD) of the papilla to the maximum point of deformation, and the other toward the interior (IntD) on the GFP image were measured, respectively.

## Oryzalin treatment

To avoid contact of pollen grains with liquid, we performed local applications of oryzalin (Chemical service, Supelco) as described (*Sassi et al., 2014*), using lanolin pasted around the style, just under the stigmatic cells, at 833 µg/mL (DMSO), for 4 hr at 21˚C. Oryzalin-treated pistils were pollinated with mature WT pollen and 1 hr after pollination observed under SEM.

## Cell wall composition analyses

XyG composition of stigmas was performed following the oligosaccharide fingerprinting set up by *Lerouxel et al., 2002*. 10 stigmas per genotype were dissected and kept in 96% ethanol. After ethanol removal, XyG oligosaccharides were generated by treating samples with endoglucanase in 50 mM sodium acetate buffer, pH 5, overnight at 37˚C. Matrix-assisted laser-desorption ionization time of flight mass spectrometry of the XyG oligosaccharides was recorded with a MALDI/TOF Bruker Reflex III using super-DHB (9:1 mixture of 2,5-dihydroxy-benzoic acid and 2-hydroxy-5-methoxy-benzoic acid; Sigma-Aldrich, sigmaaldrich.com) as matrix. Alcohol insoluble residues were obtained from around 3 to 5 mg of fresh pistils. They were collected for analysis and fixed in 96% ethanol before incubation for 30 min at 70˚C. The pellet was then washed twice with 96% ethanol and twice with acetone. The remaining pellet is called alcohol insoluble residues (AIR) and was dried in a fume hood overnight at room temperature. Cell wall monosaccharide content of the non-cellulosic fraction was determined by hydrolysis of all the AIR obtained from the pistils with 2 M TFA for 1 hr at 120˚C. After cooling and centrifugation, the supernatant was dried under a vacuum, resuspended in 200 µl of water and retained for analysis. To obtain the glucose content of the crystalline cellulose fraction, the TFA-insoluble pellet was further hydrolysed with 72% (v/v) sulfuric acid for 1 hr at room temperature. The sulfuric acid was then diluted to 1 M with water and the samples incubated at 100˚C for 3 hr. All samples were filtered using a 20-µm filter caps, and quantified by HPAEC-PAD on a Dionex ICS-5000 instrument (ThermoFisher Scientific) as described (*Fang et al., 2016*).

## Statistical analysis

Graph and statistics were obtained with R software or Excel. Statistical tests performed are specified in figure legends.

## Acknowledgements

We thank O Hamant for critical reading of the manuscript and fruitful discussion, A Boudaoud and V Battu for advice on AFM experiments, the Sice and MechanoDevo team members for discussion, P Bolland, A Lacroix and J Berger for plant care and the PLATIM imaging facility of the SFR Biosciences Gerland-Lyon Sud. We thank the Bordeaux Imaging Centre especially L Brocard and B Batailler

for TEM microscopy. We also thank the BioMeca society and Pascale Milani for the AFM measurements. LR was funded by a fellowship from the French Ministry of Higher Education and Research. The work at Laboratoire de Reproduction et Développement des Plantes was supported by Grant ANR-14-CE11-0021 and the one carried out at Institut Jean-Pierre Bourgin by Saclay Plant Sciences-SPS (ANR-17-EUR-0007), ANR-11-BTBR-444 0006-BFF and by Bio-Based Industries Joint Undertaking No 745012, GRACE.

# Additional information

## Funding

| Funder | Grant reference number | Author |
|---|---|---|
| PhD fellowship French Ministry of Higher Education and Research | | Lucie Riglet |
| Agence Nationale de la Recherche | ANR-14-CE11-0021 | Lucie Riglet<br>Frédérique Rozier<br>Chie Kodera<br>Isabelle Fobis-Loisy<br>Thierry Gaude |
| Agence Nationale de la Recherche | ANR-17-EUR-0007 | Julien Sechet |
| Agence Nationale de la Recherche | ANR-11-BTBR-444 0006-BFF | Julien Sechet |
| Bio-Based Industries Joint Undertaking | N° 745012 GRACE | Julien Sechet |

The funders had no role in study design, data collection and interpretation, or the decision to submit the work for publication.

## Author contributions

Lucie Riglet, Conceptualization, Data curation, Formal analysis, Validation, Investigation, Visualization, Methodology, Writing - original draft; Frédérique Rozier, Chie Kodera, Resources, Investigation, Methodology; Simone Bovio, Formal analysis, Investigation, Methodology; Julien Sechet, Formal analysis, Funding acquisition, Investigation, Methodology; Isabelle Fobis-Loisy, Conceptualization, Supervision, Visualization; Thierry Gaude, Conceptualization, Data curation, Formal analysis, Supervision, Funding acquisition, Validation, Visualization, Writing - original draft, Project administration, Writing - review and editing

## Author ORCIDs

Julien Sechet http://orcid.org/0000-0001-8398-4743
Thierry Gaude https://orcid.org/0000-0002-0804-9445

## Decision letter and Author response

Decision letter https://doi.org/10.7554/eLife.57282.sa1
Author response https://doi.org/10.7554/eLife.57282.sa2

# Additional files

## Supplementary files

• Transparent reporting form

## Data availability

All data generated or analysed during this study are included in the manuscript and supporting files. Source data files have been provided for Figures 1, 2, 4, 5 and 6 to 5 and Figure 5—figure supplements.

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
