## [Decision Letter]

**Acceptance summary:**

The directed growth of pollen tubes through the cell walls of stigma papillae cells on their way to the transmitting tract is a fascinating process that has been difficult to dissect. Riglet et al. tested the hypothesis that mechanical properties of the stigma papillae cell wall regulate the growth of the pollen tube on the stigma. They demonstrate that control over mechanical anisotropy mediated by the microtubule cytoskeleton (rather than overall mechanical stiffness of papillae cells) has a major impact on the growth and directionality of the pollen tubes in Arabidopsis.

**Decision letter after peer review:**

Thank you for submitting your article "Mechanical properties of the stigmatic cell wall mediate pollen tube path in Arabidopsis" for consideration by *eLife*. Your article has been reviewed by three peer reviewers, including Sheila McCormick as the Reviewing Editor and Reviewer #1, and the evaluation has been overseen by Christian Hardtke as the Senior Editor. The following individuals involved in review of your submission have agreed to reveal their identity: Arun Sampathkumar (Reviewer #2); Sharon A Kessler (Reviewer #3).

The reviewers have discussed the reviews with one another and the Reviewing Editor has drafted this decision to help you prepare a revised submission.

Summary:

Mechanical cues generated internally or externally impact growth-related processes across kingdoms. It is known that overall stiffness of the extracellular matrix impacts the growth and proliferation of different cell types (such as in cancer cells). The relative contribution of mechanical anisotropy in such processes is less clear.

The directed growth of pollen tubes through the cell walls of stigma papillae cells on their way to the transmitting tract is a fascinating process that has been difficult to dissect. In this manuscript, the authors tested the hypothesis that mechanical properties of the stigma papillae cell wall regulate the growth of the pollen tube on the stigma. This hypothesis is based on the observation that pollen tubes tend to coil around papillae cells in older stigmas and that papillae aging is associated with altered cytoskeletal arrangements, with cortical microtubules arranged more isotropically in older stigmas than in younger stigmas. In agreement with the aging stigma data, a mutant altered in microtubule dynamics, *ktn1-5*, also displayed pollen tube coiling when pollinated with wild-type pollen.

Since cortical microtubules are associated with cellulose synthesis, the authors tested whether the stiffness and cell wall composition in mutant and aging papillae was associated with pollen tube coiling and found that softer cell walls and isotropic arrangements of cellulose microfibrils in aging and *ktn1-5* papillae can lead to faster pollen tube growth through the papillae wall and loss of directionality. These results support the idea that it is control over mechanical anisotropy mediated by the microtubule cytoskeleton (rather than overall mechanical stiffness) of papillae cells has a major impact on the growth and directionality of the pollen tubes in Arabidopsis.

Overall, this is very exciting work that makes a significant contribution to the plant reproduction field. However, we have some concerns as to how some of the data is presented and interpreted. Please see detailed comments below:

Essential revisions:

We are concerned about the mechanical measurements recorded by the AFM. The AFM data makes a compelling case for wall softening in stage 15 and *ktn1-5* papillae, however it is not clear whether cell wall stiffness was measured along the papillae.

The flexible nature of the papillae cells that are only immobilized at the base when probed with an indenter could contribute to the differences in the measurements made. The authors make very small deformations, however, most measurements performed to date in plants are on rigid and relatively immobile cell and tissue types. As the most exciting finding of this study is that the difference in the growth of pollen tubes is due to modified anisotropy rather than the stiffness, this needs to be addressed.

From the SEMs throughout the paper, it seems that pollen tubes tend to adhere to either the tip of the papilla cell or just behind the tip and none of the coiling tubes shown in the figures have pollen grains directly on the tip. Is the amount of turning associated with where the pollen grain adheres (i.e. in wild-type, stage 13, did the 1/2 turn pollen tubes germinate at a different position on the papillae?) Are there differences in wall stiffness along the length of papillae cell (i.e., tip vs. flank) that could contribute to the coiling phenotypes?

It should also be noted that the measured indentation modulus using AFM mostly reflects the mechanical contribution of pectin (Zhang et al., 2019) and cellulose doesn't contribute much to this. This relates to the results of the AFM measurements of *ktn1-5* mutants that show reduced stiffness despite the biochemical analysis showing increased amounts of crystalline cellulose. If there is a role of pectin in such a process the tested *qua1* mutant should show some defects, but it doesn't. Are some of these tested candidate genes indeed expressed in papillae cells?

The Discussion mostly emphasizes only the aspect of microtubule-mediated maintenance of mechanical anisotropy (Discussion, first paragraph). However, it could also be the competence of microtubules in the papillae cells to respond and modify cell wall anisotropy to mechanical forces generated by pollen tube growth. To some extent this is mentioned in the Introduction, but not in the Discussion. Maintenance and active remodeling of wall anisotropy are two different concepts that could regulate pollen tube guidance, but data that differentiates either one of these mechanisms is not presented. Ideally, one would like to see if microtubule or cell wall organization in papillae cells change when a pollen tube is growing. Please discuss.

The false coloring of pollen tubes in the SEMs throughout the manuscript is a bit misleading since it could be interpreted by non-experts as the pollen tubes growing along the surface of the papillae cells. To increase understanding of the complexity of the system, the authors should stress that the pollen tubes enter the cell wall of the papillae cells very soon after they hydrate and germinate (according to the Kandasamy et al., 1994 paper, this occurs within 10 minutes after pollen grains land on the stigma!) We suggest including the image from Figure 5—figure supplement 1C in Figure 2 (without false coloring) to emphasize that the pollen tube is not on the surface of the papilla cell, but growing through its wall.

A second point that should be made with regards to the biology of pollen/stigma interactions is related to the aging of flowers and the *ktn1-5* mutant. It is interesting that such dramatic cytoskeletal and cellulose arrangement changes occur in the older papillae. Are there known differences between stage 13 and 15 flowers (or *ktn1-5* mutants vs. wild-type) in terms of papillae receptivity to pollen and/or ability to enter into the transmitting tract and achieve successful targeting to ovules? In other words, would pollen tube coiling be expected to have an impact on fertility or pollen competition? For example, from some of the SEMs, it appears that wild-type pollen on *ktn1-5* mutant stigmas does not hydrate as much as on wild-type stigmas. The specific softening and CMF effects on the cell wall in *ktn1-5* mutants are interesting and suggest that stage-specific KTN1 regulation could be occurring in aging papillae. The authors should comment on the possible regulation of KTN1 during stigma aging and possible implications on pollination biology in the Discussion.

The pharmacological treatment is concerning because the oryzalin concentration used for stylar applications (833 μg/mL) is around 2.4 mM, which is unusually high. Also, the microtubule network is completely missing following this treatment (Figure 4B). However, the number of turns the pollen tube makes around the papillae is much lower after oryzalin (compared to *ktn1-5*, where the microtubule network was still visible (Figure 3A and B) but in an isotropic arrangement). Did the mechanical properties and cellulose microfibril organization also change in papillae following oryzalin treatment? Did the authors try reduced concentrations of oryzalin so that the microtubule network is still visible, as in *ktn1-5*? The drug treatment result suggests that isotropic orientation of CMTs is not very important for the pollen tube coiling phenotype. Rather, the softening of papillae cell wall contributes to the pollen tube coiling phenotype as suggested in Figure 5E and F.

---

## [Author Response]

Essential revisions:We are concerned about the mechanical measurements recorded by the AFM. The AFM data makes a compelling case for wall softening in stage 15 and ktn1-5 papillae, however it is not clear whether cell wall stiffness was measured along the papillae.The flexible nature of the papillae cells that are only immobilized at the base when probed with an indenter could contribute to the differences in the measurements made. The authors make very small deformations, however, most measurements performed to date in plants are on rigid and relatively immobile cell and tissue types. As the most exciting finding of this study is that the difference in the growth of pollen tubes is due to modified anisotropy rather than the stiffness, this needs to be addressed.

The editors/reviewers are right. The pompon-like shape of the stigmatic surface confers flexibility to papilla cells that was a real concern for Young’s modulus measurements. Indeed, when applying the indenter, the papilla slightly moved and this led to erratic values. To circumvent this problem, after several trials, we found a way to immobilize the papillae while leaving the top accessible to the indenter. This procedure is now clearly stated in the Materials and methods section: “Pistils were placed straight in a 2% agar MS medium and 0.8% low-melting agarose was added up to a certain level where the papilla cells were maintained well immobilised in agarose while leaving the top accessible to the indenter. This set up allowed accurate measurements of cell wall stiffness on the dome-shaped top of papilla cells”. However, we did not succeed in measuring reproducible Young’s modulus on papillae from stigmas placed horizontally on the agarose layer.

In addition, to confirm our AFM results, a second series of experiments was carried out on Col-0 and *ktn1-5* papillae at stage 13 using another Atomic Force Microscope (JPK Nanowizard III) and another method of curve analysis based on the retract curves. The additional data (Author response image 1) confirmed that apparent young modulus is significantly lower in *ktn1-5* than in Col-0 papillae.

**Author response image 1. sa2fig1:** Stiffness differences between Col-0 and *ktn1-5* papillae at stage 13. Young’s modulus values of the papilla cell wall for Col-0 at stage 13 (N = 7 stigmas, n = 33 papillae) and *ktn1-5* (N = 7 stigmas, n = 9 papillae). Statistical differences were calculated using a T-test, ***P<0.01.

From the SEMs throughout the paper, it seems that pollen tubes tend to adhere to either the tip of the papilla cell or just behind the tip and none of the coiling tubes shown in the figures have pollen grains directly on the tip. Is the amount of turning associated with where the pollen grain adheres (i.e. in wild-type, stage 13, did the 1/2 turn pollen tubes germinate at a different position on the papillae?)

As mentioned by the reviewers, pollen adhesion often occurs at the top of papilla cells. However, we rarely saw pollen grains attached at the very tip of the papilla; we may suppose that the oval shape of dehydrated pollen grains might prevent the pollen to stay firmly attached to the tip. Why and how this preferential top localization happens and whether the landing position of the pollen on the papilla might affect the number of turns remain open questions, which we think are out of the scope of the present work. To respond more precisely to the above comment, for similar positions of the pollen on a stigma papilla, we did observe different behaviors of the pollen tube, going either straight or making ½ turn (see Author response image 2, SEM of Col-0 stigma papillae at stage 13 pollinated with Col-0 pollen). These results suggest that pollen position is not a key determinant of pollen tube behavior.

**Author response image 2. sa2fig2:** Col-0 pollen tube trajectory after attachment of the pollen to different positions on the Col-0 papilla at stage 13. Pollen tube trajectory is not dependent on the position where pollen has landed.

Are there differences in wall stiffness along the length of papillae cell (i.e., tip vs. flank) that could contribute to the coiling phenotypes?

Unfortunately, as mentioned in Response 1, due to the particular shape of papilla cells and their “pompon” organization, it was not possible to measure the Young’s modulus of the cell wall on the flank of papillae and hence to compare the cell wall stiffness between tip and flank. Based on the above observation (see Author response image 2) that pollen tube trajectory is independent of the pollen attachment position, we cannot rule out that local/heterogeneous mechanical properties of the cell wall at stage 13 may contribute to the coiling phenotype.

It should also be noted that the measured indentation modulus using AFM mostly reflects the mechanical contribution of pectin (Zhang et al., 2019) and cellulose doesn't contribute much to this. This relates to the results of the AFM measurements of ktn1-5 mutants that show reduced stiffness despite the biochemical analysis showing increased amounts of crystalline cellulose. If there is a role of pectin in such a process the tested qua1 mutant should show some defects, but it doesn't. Are some of these tested candidate genes indeed expressed in papillae cells?

We suspected that among other cell wall components, pectins could contribute to the coiling phenotype. That is the reason why we selected the *qua2.1* mutant, which is strongly affected in cell wall pectins with a 50% reduction in homogalacturonan content compared with Col-0 without observable changes in other polysaccharides (Mouille et al., 2007). As no coiling phenotype was observed in the *qua2.1* mutant, we can exclude pectins as contributing to coiling.

We agree with the reviewers that it was unclear in the first version of our manuscript whether genes of the cell wall mutants studied were actually expressed in the stigma. This is now clearly stated in the Results section and Figure 5—table supplement 1. “We selected mutants impaired in the cellulose synthase complex (*kor1.1*, *prc1* and *any1*), hemicellulose biosynthesis (*xxt1 xxt2*, *xyl1.4*) and pectin content (*qua2.1*), for which expression of the corresponding genes in stigma was confirmed (Figure 5—table supplement 1)”. Furthermore, in the Discussion, our results are now discussed with regard to the data reported in Zhang et al., 2019:

“Interestingly, Zhang et al., 2019, reported that cellulose networks largely determine in-plane mechanics whereas out-of-plane mechanics depends on both homogalacturonan (pectin) and cellulose networks. […] In light of these data, it is noteworthy that papilla cells of the *qua2.1* mutant do not induce pollen tube turns (Figure 5—figure supplement 3), which suggests that pectins play no significant role in the pollen tube phenotype and that the main factor contributing to pollen tube guidance is in-plane mechanics, and hence relies on the cellulose network”.

The Discussion mostly emphasizes only the aspect of microtubule-mediated maintenance of mechanical anisotropy (Discussion, first paragraph). However, it could also be the competence of microtubules in the papillae cells to respond and modify cell wall anisotropy to mechanical forces generated by pollen tube growth. To some extent this is mentioned in the Introduction, but not in the Discussion. Maintenance and active remodeling of wall anisotropy are two different concepts that could regulate pollen tube guidance, but data that differentiates either one of these mechanisms is not presented. Ideally, one would like to see if microtubule or cell wall organization in papillae cells change when a pollen tube is growing. Please discuss.

We absolutely agree with the reviewers’ comment and, actually, answering the question *“Ideally, one would like to see if microtubule or cell wall organization in papillae cells change when a pollen tube is growing”*, was one of our first objectives. However, this revealed technically challenging as it needed to set up a live imaging system allowing observation of the pollen tube growth on stigma papillae in the absence of a mounting medium. We finally succeeded in setting up an efficient live cell imaging system (described in the Materials and methods section). Results of this experiment are now added in the revised version and discussed accordingly; see Figure 3 and Figure 3—figure supplement 1, in the subsection “Pollen tube penetration in the papilla cell wall does not alter CMT organisation”.

Unexpectedly, we did not observe any significant change in CMT network following pollen tube invasion/growth in the papilla cell wall. This result is in contradiction with previous data reported by Samuel et al., 2011, that reported microtubule fragmentation in pollinated papillae and is discussed in the Discussion.

The false coloring of pollen tubes in the SEMs throughout the manuscript is a bit misleading since it could be interpreted by non-experts as the pollen tubes growing along the surface of the papillae cells.

To avoid misinterpretation, false coloring of pollen tubes has been removed in all figures (i.e., Figures 2, 4 and 5).

To increase understanding of the complexity of the system, the authors should stress that the pollen tubes enter the cell wall of the papillae cells very soon after they hydrate and germinate (according to the Kandasamy et al., 1994 paper, this occurs within 10 minutes after pollen grains land on the stigma!) We suggest including the image from Figure 5—figure supplement 1C in Figure 2 (without false coloring) to emphasize that the pollen tube is not on the surface of the papilla cell, but growing through its wall.

We think that reviewers might have incorrectly referred to Figure 5—figure supplement 1C as Figure 2A is very similar to Figure 5—figure supplement 1C. In addition, Figure 5—figure supplement 1C does not really show that the pollen tube grows within the papilla cell wall. We were aware of the possible misunderstanding about where the pollen tube was actually growing, in the cell wall or on the papilla surface. We think we were cautious in clearly stating from the Abstract and then in the Introduction and Results section that the pollen was growing “within the cell wall”.

– Abstract: “Pollen tubes first grow within the cell wall of the papilla cells…”

– Introduction: “In *Arabidopsis thaliana*, pollen tubes grow within the cell wall of papillae of the stigmatic epidermis, and then through the transmitting tissue of the style and ovary (Lennon and Lord, 2000)”.

– Results: subsection “Pollen tube penetration in the papilla cell wall does not alter CMT organization”.

In addition, Figure 6A-C clearly show that the pollen tube is embedded in the papilla cell wall.

A second point that should be made with regards to the biology of pollen/stigma interactions is related to the aging of flowers and the ktn1-5 mutant. It is interesting that such dramatic cytoskeletal and cellulose arrangement changes occur in the older papillae. Are there known differences between stage 13 and 15 flowers (or ktn1-5 mutants vs. wild-type) in terms of papillae receptivity to pollen and/or ability to enter into the transmitting tract and achieve successful targeting to ovules? In other words, would pollen tube coiling be expected to have an impact on fertility or pollen competition? For example, from some of the SEMs, it appears that wild-type pollen on ktn1-5 mutant stigmas does not hydrate as much as on wild-type stigmas.

Answers to these questions are now provided in the Results subsection “Impaired CMT dynamics of papillae affects pollen tube growth direction” and Figure 5—figure supplement 2. Briefly, we find the receptivity of the stigma (defined as its capacity to allow proper hydration of pollen grains and germination and growth of pollen tubes) to be unchanged between Col-0 and *ktn1-5*, although fertility (assessed from seed set) of *ktn1-5* ovules was strongly reduced. Decrease in fertility of Col-0 pistils was reported to occur only after stage 16 (Gao et al., 2018), seed set rates being not significantly different at earlier stages following anthesis. This is now discussed with regard to the potential implication of the coiling phenotype on reproductive biology.

The specific softening and CMF effects on the cell wall in ktn1-5 mutants are interesting and suggest that stage-specific KTN1 regulation could be occurring in aging papillae. The authors should comment on the possible regulation of KTN1 during stigma aging and possible implications on pollination biology in the Discussion.

See response 8 above and the Discussion.

The pharmacological treatment is concerning because the oryzalin concentration used for stylar applications (833 μg/mL) is around 2.4 mM, which is unusually high. Also, the microtubule network is completely missing following this treatment (Figure 4B). However, the number of turns the pollen tube makes around the papillae is much lower after oryzalin (compared to ktn1-5, where the microtubule network was still visible (Figure 3A and B) but in an isotropic arrangement). Did the mechanical properties and cellulose microfibril organization also change in papillae following oryzalin treatment?

Due to the constraints associated with the need to prevent artefactual hydration of pollen grains on control and oryzalin-treated stigmas, it was technically not possible to investigate the mechanical properties and cellulose microfibril organization of oryzalin-treated cell walls. It remains unclear whether such a short (4h) drug treatment might impact cellulose microfibril organization and hence cell wall mechanics. A sentence has been added in the Discussion to point out this uncertainty: “These results reveal a link between the stigmatic CMT cytoskeleton organisation and the trajectory that pollen tube takes while growing in the papilla cell wall, although the effects of oryzalin on cell wall properties remain unknown”.

Did the authors try reduced concentrations of oryzalin so that the microtubule network is still visible, as in ktn1-5? The drug treatment result suggests that isotropic orientation of CMTs is not very important for the pollen tube coiling phenotype. Rather, the softening of papillae cell wall contributes to the pollen tube coiling phenotype as suggested in Figure 5E and F.

We did try a series of concentrations and duration of oryzalin treatments. To avoid misinterpretation of the effect of oryzalin on pollen tube growth, we chose a concentration/duration of oryzalin treatment that led to a homogeneous depolymerization of microtubules. Author response image 3 shows some examples of concentration/duration combinations we tested.

**Author response image 3. sa2fig3:** Col-0 papilla cells expressing MAP65. 1-citrine after 6 or 4 hours of oryzalin local treatment. CMT depolymerization is complete and homogeneous after 4 hours of treatment at 888µg/mL. At 404 µg/mL and 666 µg/mL, CMTs are not completely destabilized.